# Effect of High-Volume Training on Psychological State and Performance in Competitive Swimmers

**DOI:** 10.3390/ijerph19137619

**Published:** 2022-06-22

**Authors:** Oussama Gaied Chortane, Sofiene Amara, Tiago M. Barbosa, Raouf Hammami, Riadh Khalifa, Sabri Gaied Chortane, Roland van den Tillaar

**Affiliations:** 1Higher Institute of Sport and Physical Education of Ksar-Said, University of La Manouba, Tunis 2010, Tunisia; oussama.gaeid@gmail.com (O.G.C.); raouf.cnmss@gmail.com (R.H.); riadhkhal@yahoo.fr (R.K.); sabrigaied1@gmail.com (S.G.C.); 2Research Unit Sports Performance, Health and Society, Higher Institute of Sport and Physical Education of Ksar-Said, University of La Manouba, Tunis 2010, Tunisia; 3Research Center in Sport, Health and Human Development, 5000-801 Vila Real, Portugal; barbosa@ipb.pt; 4Department of Sports Sciences, Instituto Politécnico de Bragança, Campus Sta., 5301-856 Bragança, Portugal; 5Laboratory of Cardio-Circulatory, Respiratory, Metabolic and Hormonal Adaptations to Muscular Exercise, Faculty of Medicine Ibn El Jazzar, Sousse 4002, Tunisia; 6Department of Sport Sciences and Physical Education, Nord University, 7600 Levanger, Norway

**Keywords:** somatic anxiety, cognitive anxiety, self-confidence, external training load

## Abstract

This study aimed to examine the effect of four weeks of aquatic high-volume training (HVT) on the psychological state (somatic, cognitive anxiety, and self-confidence) and sprint swimming performance (50 m front crawl) compared to the standard training program (moderate volume training) in competitive swimmers. Twenty-eight male competitive swimmers participated in this study and were randomly allocated into two groups: HVT group (*n* = 14; age = 16.4 ± 0.31 years) and control group that underwent the standard training program (*n* = 14; age = 16.1 ± 0.30 years). All psychological state variables and swimming performance were measured in pre and post-test. Our findings showed a significant increase in anxiety state (34.13% to 45.83%; ES = 3.26 to 3.38) and a significant decrease in self-confidence (18.43%; ES = 2.39) after four weeks of HVT, while all psychological state variables remained unchanged in the control group (*p* > 0.05). In addition, our results showed no significant enhancement in swimming performance in both groups (*p* > 0.05). The sudden increase in training mileage negatively affected the anxiety, decreasing the state of self-confidence of the participants. In addition, four weeks of high training volume are insufficient to improve swimming performance. To conclude, gradually increasing the volume of the training load may be an adequate solution to promote adaptation to the effort, thus maintaining the stability of the psychological state of swimmers. In addition, it is recommended to integrate a concurrent mental preparation program with high-volume training to monitor the psychological state of competitive swimmers.

## 1. Introduction

Optimizing swimming performance involves improving physical, psychological, and biomechanical variables [1,2,3]. More specifically, swimming coaches and swimmers design and implement aquatic and dry land training programs to optimize swimming performance [4,5]. High volume training is basic training aimed at improving aerobic capacities, (e.g., aerobic power, aerobic threshold) and consequently optimizing performance in competitive swimmers [6]. Lauren [7] showed that high volume training with 75% of total training volume performed at low intensity and 10–15% at very high intensity is an optimally recommended training for elite athletes in intense exercise sports, (e.g., swimming and Olympic rowing) to optimize specific performance. In the same context, Pugliese et al. [6] showed that high volume training (total week training volume = 12 km) could improve peak oxygen consumption (11.9 ± 4.9%) and swimming performance (400 m: −2.8 ± 1.8%; 2000 m: −3.4 ± 2.9%) in master male swimmers (age 32.3 ± 5.1 years). The planning of swimming training volume must take into account certain factors, (e.g., age, level of training, swimming specialty). In fact, the training volume can range from 4000 m up to 7000 m per session in young competitive swimmers (between 13 and 17 years old) [8]. However, the management of the psychological state during the swimming career is very important, in order to achieve optimal performance in competitive swimmers [9].

Several studies have considered that training volume can be divided into two concepts: (1) quality training which is characterized by low-volume, high-intensity training (HIT) and is aimed at improving resistance performance for swimmers and generally incorporated for swimmers specializing in short distances [8,10]. (2) Quantity training, which is characterized by high-volume, low-intensity training (HVT) and is aimed at improving respiratory capacity, and adaptations to skeletal muscle for swimmers and is generally incorporated at the start of the training season [10,11]. However, swimming coaches are required to balance training and avoid sudden increases in training volumes. Therefore, the volume of training presents an important debate and especially among adolescent swimmers. Notably, Feijen et al. [12] showed in a systematic review that the greatest swimming training volumes were recorded in adolescent (17.27 ± 5.25 h/week) and adult swimmers (26.8 ± 4.8 h/week) and this caused significant pain in the shoulders in a large number of adolescent swimmers (91.3% compared to other age groups). This is what has made the study of the effect of the sudden increase in training volume on the anxiety state of swimmers a focus of attention for many swim coaches.

Anxiety, defined as an emotional state, is characterized by physiological stimuli, associated with feelings of nervousness and worry [13]. The value of anxiety depends on the individual’s ability to respond to physical and psychological demands [13,14]. On the other hand, psychometric questionnaires are among the methods of measuring anxiety. These questionnaires are divided into two categories, the first for studies that treat anxiety as pathology, while the second category of questionnaires was used by the authors to study anxiety as a normal personality trait [15,16]. Several authors have studied the relationship between anxiety state and swimming performance [17,18]. Dalamitros et al. [18] had examined the effects of an exercise of five box jumps on 50 m breaststroke, psychological (anxiety), and physiological variables such as oxygenation of lower limb muscles and heart rate in expert and non-expert swimmers. Meanwhile, Fortes et al. [19] had investigated the relationship between the competitive anxiety state (400 m front crawl) and heart rate variability in swimmers at the Brazilian National Swimming Championships where swimmers responded to the Competitive Anxiety Questionnaire (CSAI-2R) and they explained that athletes with a high level of anxiety presented disturbances in the autonomic nervous system.

Competitive swimming training can affect the psychological state of swimmers [20]. For instance, Vacher et al. [9] showed that emotional states, (e.g., anxiety, happiness and excitement) were characterized by distinct trajectories during the training period preceding a major competition in high-level swimmers. In fact, each swim training period was characterized by variation in training volume and intensity. The question that can be asked here is, could increasing training volume affect the psychological state of young swimmers? According to the previous literature, swimming training and competition could influence the state of anxiety [9,18]. Nevertheless, to what extent can a change in anxiety state affect the swimming performance of young swimmers?

Information regarding the effect of varying training volume on psychological states in young swimmers requires further clarification. The present study was the first investigation that studied the effect of increased training volume on anxiety state and swimming performance in young competitive swimmers. We hypothesized that high-volume training could increase anxiety levels in young swimmers, and this could negatively affect swimming performance. 

## 2. Materials and Methods

### 2.1. Experimental Approach to the Problem

A randomized controlled trial was designed to address the research questions of the effect of four weeks of high-volume training on psychological state and swimming performance in competitive swimmers assigned to a high-volume training program and a standard program. All independent variables (cognitive anxiety, somatic anxiety, self-confidence, and 50 m front crawl) were measured in pre- and post-training. The present study was carried out during the sports swimming season (September to November).

### 2.2. Subjects

Twenty-eight male competitive swimmers were randomly allocated into two groups. High-volume training group (HVT group: *n* = 14, age = 16.4 ± 0.31 years; height: 173 ± 9.82 cm; body mass = 72.5 ± 5.36 kg), and control group under a standard training program (*n* = 14, age = 16.1 ± 0.30 years; height: 175 ± 9.40 cm body mass 74.6 ± 5.11 kg). An a priori power analysis (G * Power 3.1.9.3, Heinrich Heine Universität Düsseldorf, Düsseldorf, Germany) yielded a sample size of at least 14 swimmers per group to detect large effects (*d* = 0.98), assuming a power of 0.8 and alpha of 0.05. All swimmers were aware of the training loads for both groups and the study design and swam under the instructions of the same coaches. The best performance time of the best swimmer in 50 m front crawl was 26.10 s. All competitive swimmers had more than three years of experience in national level training and competition. All participants and parents read and signed a written informed consent (potential benefits and risks). This study was approved by an institutional review board of the Higher Institute of Sport and Physical Education of Ksar Said, University of Manouba, Tunisia (Research Unit of Sports Performance, Health and Society, UR17JS01), and was established according to the last declaration of Helsinki.

### 2.3. Aquatic Swimming Training

Swimming training sessions and tests were performed in a 25 m indoor pool with 27.1 °C and 25.9 °C water and air temperatures, respectively, and 64% of relative humidity. Training programs are designed according to the literature [11,21]. The programs took 4 weeks and included: (1) aerobic training: warm-up, technical drills, i.e., 6 × 200 m, 4 × 400 m; at 50% to 75% of maximal heart rate (HRmax); (2) High-intensity interval training (HIIT): aquatic resistance training, i.e., 5 × 100 m, 10 × 50 m; at 75% to 85% of HRmax; (3) high-intensity training (HIT): sprint training; i.e., 8 × 25 m, 2 × 6 × (15 m) at 85% to 95% of HRmax.

The HVL group underwent a high-volume training (total distance = 132.40 km): 75% of aerobic training, 15% of HIIT, and 10% of HIT. Conversely, the control group carried out the standard training program characterized by moderate volume training (total distance = 117.80 km): 75% of aerobic training, 15% of HIIT, and 10% of HIT (Table 1, Figure 1). Rating of perceived exertion load (RPE load) was calculated by multiplying the swimmer’s rating of perceived exertion (RPE, scale 1–10) after 30 min of each session by the training volume (duration) of the session [22,23,24] (Table 1).

### 2.4. Psychological State Measurements

The assessment of the psychological state of the swimmers was assessed on the 50 m front crawl test day (before and after the four weeks of training; one hour before the all-out sprint test) by a CSAI-2R questionnaire [15]. The 17-item CSAI-2R questionnaire is divided into three blocks. The first is to assess cognitive anxiety, it is made up of 5 items with an intensity between 5 and 20. However, somatic anxiety evolves in the patient. The second block with 7 items and intensity between 7 and 28, while the self-confidence evolves during the third block with 5 items and intensity between 5 and 20. The swimmer was asked to answer the questionnaire, and their answer is evaluated on a four-point scale: (1) “Not at all”, (2) “A little”, (3) “Moderately” and (4) “Very well.” In the present study, we used the French version, which was validated by Martinent et al. [25]. The score for each block was obtained by adding up, dividing by the number of items, and multiplying by 10. The intraclass correlation coefficient (ICC) of all psychological state variables for the pre-test and post-test reliability ranged between 0.81 and 0.87.

### 2.5. Swimming Performance Test

The swimming performance test was performed in the morning, noted in seconds, and was measured by two expert timekeepers per stopwatch (SEIKO S120-4030, Tokyo, Japan). The 50 m front crawl performance (T 50 m) was performed by diving start, and all swimmers were invited to perform a warm-up before the starting of the test composed of aerobic training (600 m front crawl) progressive sprint (4 × 50 m) and technical training, (i.e., diving, turn and front crawl stroke) [1]. The intraclass correlation coefficient (ICC) for the pre-test and post-test reliability was 0.91.

### 2.6. Statistical Analysis

Values are presented as mean ± SD. Relative change over time, (i.e., delta change) is reported for each variable between pre- and post-test. Normality and sphericity of the data sets were verified using the Kolmogorov–Smirnov and Mauchly tests, respectively. The intraclass correlation coefficient (ICC) was selected to calculate the reliability of the measurements between test and posttest [26]. ANOVA with repeated measures was used to determine the effect of time, group, and time × group interaction (group and time factors). Repeated measures ANOVA (time factor) was used to determine the changes in psychological state variables and swimming performance within the group if the group × time interaction reached the level of significance. The difference in volume training between both groups was calculated by the Student’s *t*-test. The effect size (ES) was calculated by converting a partial eta squared to Cohen’s d [27], where the ES can be classified as small (0.00 ≤ d < 0.2), moderate (0.2 ≤ d < 0.5), and large (d ≥ 0.5). The level of significance was established at *p* ≤ 0.05. Data analyzes were performed using SPSS version 26 for Windows (SPSS Inc., Chicago, IL, USA).

## 3. Results

A significant difference was recorded in training volume between the two groups (*p* < 0.05). No statistically significant differences were shown in the baseline values between the groups in all anthropometric variables, swimming experience training, psychological variables, and swimming performance (50 m front crawl) (*p* > 0.05). 

A significant effect of the time (*p* < 0.0, 1.46 < ES < 2.08), and the group (*p* < 0.05, 0.71 < ES < 2.28) was found in all psychological variables. A significant increase in somatic anxiety (34.13%, *p* < 0.001, ES = 3.38 [large]), in cognitive anxiety (45.83%, *p* < 0.001, ES = 3.26 [large]) and a significant decrease in self-confidence (18.43%, *p* < 0.001, ES = 2.39 [large]) in HVT group after four weeks of the intervention period were observed. No significant changes in all psychological state variables were observed in the control group (0.088 < *p* < 0.323) (Table 2). The 50 m front crawl performance remained unchanged in both groups after four weeks of swimming training (0.222 < *p* < 0.269) (Table 2).

## 4. Discussion

The aim of the present study was to investigate the effect of high-volume training on the psychological state and swimming performance. The main findings were a significant increase in the anxiety state and a significant decrease in self-confidence in the experimental group, with no changes in the control group. 

A high volume swimming training for four weeks increased the anxiety states (34.13% to 45.83%). More specifically, the sudden increase in the training volume could influence the physical state of the athlete by increasing the RPE which can increase somatic and cognitive anxiety in swimmers [28,29]. In addition, maladjustment, (i.e., decreased muscle performance, chronic fatigue) with the new high-volume training could direct the athlete to a state of overtraining, causing psychic fatigue and increased cognitive anxiety in competitive athletes [29]. As shown in Table 2 the self-confidence state was decreased in the HVL group (18.43%). The increase in cognitive and somatic anxiety after four weeks of training with a high-volume load could be the cause of this decrease in self-confidence in competitive swimmers. In fact, the inability of swimmers to keep up with the increased training load imposed could negatively affect the state of well-being and self-confidence in athletes [30]. In addition, the disturbance in psychophysiological variables due to overtraining (muscle fatigue, increased heart rate at rest, and mental fatigue) are the main factors that can decrease self-confidence in competitive athletes [31]. More specifically, the sudden increase in training load could cause significant pain in the shoulders of swimmers and consequently thickening of the tendon [12], which causes a state of psychological instability in adolescent swimmers. On the other hand, the psychological state remains stable after four weeks of usual training in control group swimmers. The adaptation period (usual training) and the training load adequate to the swimmers’ abilities could be the factors that explain these results [32]. In fact, the previous idea confirmed that the inadequacy of a new level of training volume, and the sudden increase in training load affected the psychological state of competitive swimmers.

Our results showed that 50 m front crawl performance remained unchanged after four weeks in both groups. These results are in line with Faude et al. [11], who showed that four weeks (six sessions per week) of high-volume training (64.78 ± 23.7 km) did not improve the performance of 100 and 400 m front crawls (*p* > 0.05) in competitive swimmers (age = 16.6 ± 1.4 years). Similar findings stated in several previous studies show that to optimize swimming performance, it is recommended that the intervention periods should last more than four weeks of training [8,19]. For instance, Amara et al. [8] noted that a period of eight weeks of training with a total training volume of 201 km could optimize the performance of 100 m butterfly (3.56%) in young competitive male swimmers (age = 14.10 ± 0.30 years).

In another context, the lack of a taper period in this present study may be one of the factors behind the performance stagnation in the two groups. Several studies have shown the importance of the existence of a taper period after the intervention period to promote the transfer of gains, which is because of training during the intervention period [1,2]. The taper period is characterized by a reduction in training volume while the intensity should remain high.

To summarize, a sudden increase in the training volume had a negative effect on the state of somatic and cognitive anxiety, and consequently, a decrease in the state of self-confidence appeared. Additionally, four weeks of high-volume training was insufficient to improve sprint swimming performance. The present study provides some practical recommendations. Coaches should gradually increase the training volume to promote adaptation to the effort. Thus, maintaining the stability of the psychological state of swimmers. A steep increase in the volume as reported in the HVT group, conversely, should be discouraged as this leads to increased anxiety. Yet, regardless of the intervention program, there is no change in the swimming performance. As such, four weeks of training with no taper period is not enough for a meaningful improvement in sprint performance. That said, if a steep increase in volume is necessary, it is recommended to integrate a concurrent mental preparation program to monitor the psychological state of competitive swimmers. 

One can note a few limitations to this research. It is unclear how high-volume training might affect other psychophysiological variables, (e.g., heart rate, perception of fatigue) that might play a mediator role between swim performance and anxiety. In addition, only male swimmers were recruited. Future studies must also include female swimmers to have a better understanding of how the female gender copes with HVT. In addition, it is important to study the training effect reported here on other variables that determine the psychophysiological state, (e.g., psychological and physical fatigue).

## 5. Conclusions

Our findings revealed that four weeks of aquatic high-volume training negatively affects the psychological state of national level swimmers. The sudden increase in training load leads to an increased state of somatic and cognitive anxiety. In turn, these decreased the state of self-confidence. In addition, four weeks of high-volume training is insufficient to improve the sprint swimming performance. 

## Figures and Tables

**Figure 1 ijerph-19-07619-f001:**
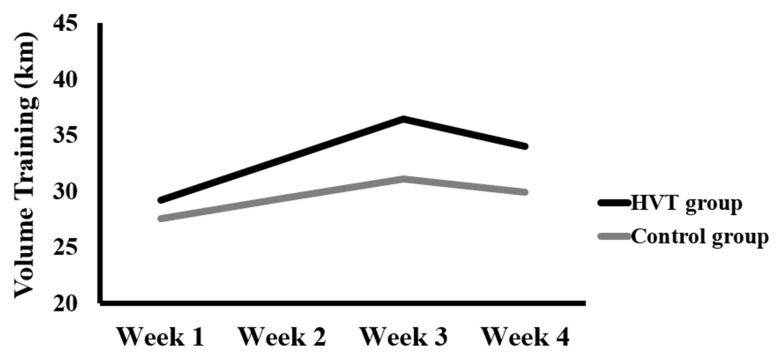
Volume training during four weeks of intervention period in both groups. HVT: High volume training.

**Table 1 ijerph-19-07619-t001:** Mean volume training and rating perceived exertion load in both groups during 4 weeks of training.

Weeks	Groups	Mean Distance (m)	Mean Duration (min)	Mean RPE (a.u)	Mean RPE Load (a.u)
Week 1	HVT group	4866.70 ± 103.28	94.67 ± 3.26	4.53 ± 0.41	430.10 ± 51.75
	Control group	4583.30 ± 98.32	87.00 ± 4.59	4.47 ± 0.32	388.92 ± 34.02
Week 2	HVT group	5466.7 ± 103.28	111.50 ± 1.94	6.62 ± 0.38	739 ± 69.30
	Control group	4883.30 ± 98.32	100.50 ± 2.53	5.15 ± 0.20	517.98 ± 33.44
Week 3	HVT group	6066.7 ± 103.28	122.17 ± 2	8.12 ± 0.43	991.93 ± 61.27
	Control group	5183.30 ± 98.32	109.50 ± 2.67	5.35 ± 0.32	585.9 ± 38.25
Week 4	HVT group	5666.7 ± 103.28	106.00 ± 1.76	7.35 ± 0.21	779.10 ± 28.89
	Control group	4983.30 ± 98.32	103.33 ± 6.65	4.85 ± 0.11	501.10 ± 32.83

**Table 2 ijerph-19-07619-t002:** Changes in psychological state and swimming performance between pre- and post-test after an intervention period of 4 weeks in high volume training (HVT) and a control group carrying out the standard training program.

Variables	Groups	Pretest	Posttest	*p*-Value	Effect [95% CI]	Delta Change (%)	ES
Somatic anxiety	HVT group	15.00 ± 1.18	20.12 ± 1.94	<0.001	−5.18 [−6.61 to −3.75]	34.13	3.38 [large]
	Control group	14.18 ± 1.25	14.73 ± 1.27	0.323	−0.55 [−1.67 to 0.58]	3.88	0.45 [moderate]
Cognitive anxiety	HVT group	10.91 ± 1.92	15.91 ± 1.22	<0.001	−5 [−6.43 to −3.57]	45.83	3.26 [large
	Control group	10.91 ± 1.76	12.09 ± 1.30	0.088	−1.18 [−2.56 to 0.19]	10.82	0.81 [large]
Self-confidence	HVT group	14.27 ± 1.35	11.64 ± 0.92	<0.001	2.64 [1.61 to 3.67]	18.43	2.39 [large]
	Control group	14.27 ± 1.62	13.36 ± 1.12	0.141	0.91 [−0.33 to 2.15]	6.38	0.69 [moderate]
T 50 m (s)	HVT group	27.36 ± 0.89	26.87 ± 0.92	0.222	0.49 [−0.32 to 1.29]	1.80	0.57 [moderate]
	Control group	27.33 ± 0.75	26.96 ± 0.75	0.269	0.37 [−0.30 to 1.03]	1.35	0.51 [moderate]

HVT: High volume training; ES: effect size.

## Data Availability

The data presented in this study are available on reasonable re-quest from the corresponding author.

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
