# Peer review of "Effect of High-Volume Training on Psychological State and Performance in Competitive Swimmers"

_ijerph, 2022, doi:10.3390/ijerph19137619_

Round 1

Reviewer 1 Report

Based on the previous peer review comments, the manuscript is easy to read. The limitations of the present study are also well described, which should be acceptable for acceptance. If I may offer some advice to the authors, it is known that negative psychological states such as loss of self-confidence and increased anxiety can cause changes in psychophysiology. For example, when a person is exposed to mental stress for a long time, the secretion of stress substances such as amylase and cortisol, and immune substances such as immunoglobulin A in saliva fluctuates. In the case of an overstimulation of the body by increased training, as in the authors' study, it is expected that the musculoskeletal response would be different from that of a simple presentation of mental stress, since the musculoskeletal response is also involved. However, we hope that the authors will consider measuring this stressor in their future work.

Author Response

Dear reviewer,

tank you for the feedback. We will in the future measure this stressor.

Reviewer 2 Report

Few minor comments:

1. Font size in tables 1 and 2 and alignment of the title in tables 1 and 2 looks different.

2. Checking for syntax and grammar in Lines 192-196 is necessary (verb is missing). Please, capitalize first letter after a dead stop (Ln193).

3. Journal titles in references appear either fully written or abbreviated, this should be unified. 

There are no further comments.

Author Response

Few minor comments:

  1. Font size in tables 1 and 2 and alignment of the title in tables 1 and 2 looks different.

That is correct. The alignment is now corrected. The font size in the table is different due to the fact that otherwise table 2 does not fit on the page. We can also make the font size of table similar to tabe 2 if the reviewer wants.

  1. Checking for syntax and grammar in Lines 192-196 is necessary (verb is missing). Please, capitalize first letter after a dead stop (Ln193).

Thank you for this comment. We have changed the text and included verb to the sentences.

  1. Journal titles in references appear either fully written or abbreviated, this should be unified. 

We have abbreviated all journal titles now for all references.

This manuscript is a resubmission of an earlier submission. The following is a list of the peer review reports and author responses from that submission.

Round 1

Reviewer 1 Report

This study showed that four weeks of over-intensity training on junior swimmers did not improve their athletic performance, but did increase anxiety and decrease confidence. The findings of this study can be said to be an exploration of the psychological aspects of why mental burnout often occurs in athletes. This is an important finding when considering means of preventing burnout among athletes in the field of sports coaching.

However, the authors do not give sufficient consideration, especially in their discussion. Furthermore, it is questionable whether measures unique to competition can be adequately addressed without describing the differences between the cases of increased anxiety and loss of confidence described in general psychology and those in sports psychology. In addition to this, the authors use a variety of inappropriate word usage and contextual settings, which may mislead the reader.

Therefore, the authors would like to be corrected based on the following comments.

[Introduction]

From lines 61 to 64: Authors stated as However, Fortes et al…’,     but this could be replaced as Meanwhile, Fortes et al….

From lines 64 to 66: The statement The same authors [20] should be and they explained that… because the description is connected to the context of the previous sentence.

[Results]

Line 171: authors describe the classification of the effect size using Cohens d. They set d > 0.5 as the large, while previous studies have set it to d > 0.8. Why do the authors change this value? It is necessary to describe the rationale the value changed.    

[Discussion]

When reading the entire discussion, the reader may have the impression that the flow of the text describes the results obtained by the authors, followed by the findings of previous studies. The discussion should consider what can be said from the authors' results based on their findings, what the authors' results mean when compared to the views of previous studies and the authors' results, and why these results were obtained. However, this discussion is insufficient, so we would like to see a more detailed discussion.

For more detailed comments, please refer to the following description.

Line 192: The sentence The mean findings… should be The main findings…

Lines from 195 to 196: Authors stated as Aquatic training…. Which figure or table does a reader see?

Lines from 201 to 202: The sentences here are perceived as having a contextual connection to the previous sentences. However, the authors broke the line here. Why are you doing this? The reader may think that a line break switches to a sentence with a different main idea.

Line 202: The sentence Our findings showed… should be As shown in Table_ (or Figure_), we showed….

Lines from 202 to 209: The authors report that increased training volume leads to a loss of confidence and an increase in anxiety in athletes and present the views of previous studies. However, it is not clear what the authors can say about their own results based on this view.

Lines from 210 to 215: This paragraph could be read as a continuation of the previous paragraph. Then, the first sentence would be in the context of "On the other hand, the group that trained normally for 4 weeks...".

Line 220: The sentence In the same context, several previous studies… should replace Similar findings are obtained in several previous studies show that…

Lines from 244 to 250: Authors state the limitation of their study. Especially, they state psychophysiological variables like heart rate variables, perception of fatigue, and so on, which estimate the activity state of the autonomic nervous system. But authors do not describe the relationship between the mental state and the activity of the autonomic activity in this part with referencing previous studies.  Further, the authors also state the change in the mental conditions of female swimmers in the lines from 247 to 248. The authors should describe the previous studies that studied female athletes mental conditions other than swimmers.

Author Response

See response file

Reviewer 2 Report

The study is well-written, grammar and syntax are in general acceptable, still English language proof is recommended for minor corrections. The study has merits, it is important to understand the impact of heavy training load particularly in young swimmers. Nevertheless, I believe the manuscript could be improved.

The authors should first establish the background of the training volume in age-group swimmers, how training volume is related to performance and how necessary is to increase it. Then, the connection of training volume with any potential negative effects should be also mentioned and discussed.

In addition, why did the authors choose to measure swimming performance with a 50m swim test? 50m is a sprint event in swimming, it is reasonable to think therefore that training volume will not significantly affect performance in a sprint event. Moreover, mean difference in training mileage between the two groups was 12.3%, is this enough to induce significant differences in performance within a mesocylce? This is an important aspect and should be considered.  

Also, the authors should better emphasize that this study examines short-term effects of one mesocycle, however long-term effects are not known. This would be an interesting aspect to examine regarding both psychological state and performance capacity.

Abstract

Please, include the results from the swim test for both groups.

Introduction

Given that this study examines the effects of increased training volume on performance a brief literature review on this is necessary. How training volume affects performance in swimming and how beneficial is for improving performance? Then, the connection of anxiety state and swimming performance should be better addressed. The authors refer to some papers in the literature, however the results and conclusions from these studies are neither mentioned nor discussed. I see no point to include in the manuscript papers discussing the positive effects of recreational swimming (Ln67-70), it seems out of this manuscript’s scope. Why do the authors hypothesize that training volume increase would increase anxiety? what is the theoretical background and/or previous research supporting this?

Methods

Some important aspects in the study design that need to be addressed: The participants were from the same swimming group and swim under the instructions of the same coach? Were the swimmers aware of the training loads for the two groups and of the study desing? Did they swim next to each other?

Besides training volume training content (in terms of swimming stroke) was the same for both groups?

Please, specify the meaning of a standard training program

Results

Was there any difference in training volume between the two groups? The authors mention that a comparison using student-t test was performed, however the results are not included.

What is the reason to include two different calculations for effect size?

Discussion

The applied training mileage is only mentioned, but not discussed at all. Using previous research to support it, how common is the training load at this age-group, is it considered high/low?  

It is recommended to discuss in more details the increase in training volume, which is is characterized by the authors as sudden in the HTV group. Training volume increased by 300m (~6%) per week for the control group until the 3rd week and by 600m (~11%) for the HTV group. Even in the HTV group this increase does not seem very large, particularly in early season when the measurements were performed. Analysis here should also include previous research examining the effects of changes in training volume on swimming performance. Terms like steep or gradual increase should be better explained and analysed. It is also important to know the athlete’s maximal training mileage throughout the season.

Also, it would be useful to see the training volume increase in previous periods. If all swimmers were used to have a weekly increase of about 6%, then an increase of 10-12% is quite possible to have unfavourable effects on their overall well-being.

Again, why to expect performance improvement in a sprint event with interventions only in training volume? Were there any changes in aerobic capacity or aerobic/anaerobic thresholds between the two groups? A simple comparison in 50m all-out swim is not enough to draw conclusions regarding the athletes’ performance capacity. Volume training most likely will not improve sprint performance.

Author Response

See response file

Round 2

Reviewer 2 Report

In general, the corrections in the manuscript are in the right direction, however, I feel that several additional corrections could have been made.

The connection of training volume with swimming performance in adolescent competitive swimmers is not still discussed adequately. The reader should understand the practical meaning of training volume in swimming practice. How beneficial or harmful can be? This is a matter of cost and benefit. Typically, larger training volume in swimming results in better performance, particularly in young swimmers. However, as the authors correctly address in the manuscript, sudden increase in swimming mileage can have unfavourable effects, probably resulting in burn out and early cessation of the athletic career. This issue should be discussed and analysed in both the introduction and the discussion.

Again, a concern regarding the study design. Does the fact that both groups were aware of each other’s training volume may have influenced psychological measurements? It shouldn’t be very attractive to see the swimmers next to you swim half of the distance you have to swim.

Citation in Ln45 seems not correct, please check, and correct accordingly.

I suggest adding summarized results for total training volume (4 weeks together) and to include effect size statistics between the two groups in Ln172 (with mean values).

It is recommended to better highlight that HVT training group had a 2-fold increase in volume compared to the control group. This is an easy-to-follow quantification of the increase rate.

I understand that the authors examined the effects of HVT training only on sprint performance (and not on other performance indicators), but this must be addressed in the manuscript, and it must be considered when interpreting the results. I didn’t find any reference to this aspect in the text.